# Pevonedistat Inhibits SOX2 Expression and Sphere Formation but Also Drives the Induction of Terminal Differentiation Markers and Apoptosis within Arsenite-Transformed Urothelial Cells

**DOI:** 10.3390/ijms24119149

**Published:** 2023-05-23

**Authors:** Aaron A. Mehus, Madison Jones, Mason Trahan, Kaija Kinnunen, Kaitlyn Berwald, Becker Lindner, Sarmad Al-Marsoummi, Xu Dong Zhou, Scott H. Garrett, Donald A. Sens, Mary Ann Sens, Seema Somji

**Affiliations:** Department of Pathology, School of Medicine and Health Sciences, University of North Dakota, Grand Forks, ND 58202, USA; madison.gail.jones@und.edu (M.J.); mason.trahan@und.edu (M.T.); kaija.kinnunen@und.edu (K.K.); kaitlyn.bewald@und.edu (K.B.); becker.lindner@und.edu (B.L.); sarmad.al.marsoummi@und.edu (S.A.-M.); xudong.zhou@und.edu (X.D.Z.); scott.garrett@med.und.edu (S.H.G.); donald.sens@med.und.edu (D.A.S.); mary.sens@und.edu (M.A.S.); seema.somji@und.edu (S.S.)

**Keywords:** stem cells, terminal differentiation, basal, urothelial carcinoma, arsenite

## Abstract

Urothelial cancer (UC) is a common malignancy and its development is associated with arsenic exposure. Around 25% of diagnosed UC cases are muscle invasive (MIUC) and are frequently associated with squamous differentiation. These patients commonly develop cisplatin (CIS) resistance and have poor prognosis. SOX2 expression is correlated to reduced overall and disease-free survival in UC. SOX2 drives malignant stemness and proliferation in UC cells and is associated with development of CIS resistance. Using quantitative proteomics, we identified that SOX2 was overexpressed in three arsenite (As^3+^)-transformed UROtsa cell lines. We hypothesized that inhibition of SOX2 would reduce stemness and increase sensitivity to CIS in the As^3+^-transformed cells. Pevonedistat (PVD) is a neddylation inhibitor and is a potent inhibitor of SOX2. We treated non-transformed parent and As^3+^-transformed cells with PVD, CIS, or in combination and monitored cell growth, sphere forming abilities, apoptosis, and gene/protein expression. PVD treatment alone caused morphological changes, reduced cell growth, attenuated sphere formation, induced apoptosis, and elevated the expression of terminal differentiation markers. However, the combined treatment of PVD with CIS significantly elevated the expression of terminal differentiation markers and eventually led to more cell death than either solo treatment. Aside from a reduced proliferation rate, these effects were not seen in the parent. Further research is needed to explore the potential use of PVD with CIS as a differentiation therapy or alternative treatment for MIUC tumors that may have become resistant to CIS.

## 1. Introduction

Bladder cancer is the fourth most common cancer in men and according to the American cancer society, about 82,290 new cases will be diagnosed in 2023 with approximately 16,710 deaths occurring from the disease [1]. Urothelial cancers (UC) are the most common type of bladder cancer and they are of two types; non-muscle invasive UC (NMIUC) and muscle invasive UC (MIUC). These two types of UCs are subtyped into various groups with the basal and luminal subtype being the most prevalent [2,3]. The basal subtype of UC is more aggressive then the luminal subtype and they often have a poorer outcome compared to the luminal subtype [3,4]. Some of the basal MIUC’s exhibit areas of squamous differentiation that may be associated with increase resistance to chemotherapeutic agents [2,5].

Approximately 25% of the patients with UC have a muscle invasive disease and the treatment is a combination chemotherapy with methotrexate, vinblastine, doxorubicin and cisplatin (CIS). Recently, gemcitabine along with CIS has demonstrated similar outcomes with lower side effects [6,7,8,9]. Unfortunately, the relapse rate following treatment is high (30–54%) with a majority of the individuals succumbing to the disease and this has been attributed to resistance to the treatment [8,10,11]. The proposed mechanisms involved in resistance are varied and include increase expression of multidrug resistance genes, enhanced DNA repair capacity, and resistance to apoptosis [12]. There is an emerging role of cancer stem cells in UC where it is believed they contribute to the development of resistance to chemotherapeutic drugs [13,14].

The sex-determining region Y (SRY) associated high-mobility group (HMG) Box (SOX) transcription factors are pluripotency associated stem cell factors involved in various process during embryogenesis and are aberrantly expressed in many cancers [15]. One of the family members SOX2 is overexpressed in cancers of the breast, lung, ovary, colon and prostate and its expression is positively correlated with poor patient outcomes [16,17]. In UC, the expression of SOX2 varies according to the grade and stage of the disease with higher expression in high-grade NMIUC and MIUC compared to low-grade NMIUC and it is proposed that it plays a role in tumor maintenance and progression of the disease [18]. With its higher expression in bladder cancer, SOX2 may be a potential target for bladder cancer therapy.

Yin et al. have shown that Pevonedistat (PVD, MLN4924), a small molecule inhibitor of neddylation down regulates the expression of SOX2 by inactivating the ligase FBXW2 E3 [19]. This results in the accumulation of the SOX2 transcriptional repressor MSX2 resulting in the repression of SOX expression. In addition, the study shows that down regulation of SOX2 sensitizes breast cancer cells to tamoxifen. Another study has shown that PVD inhibits the growth of UC cell lines and in-vivo tumor xenografts [20]. Combined, these studies suggest that inhibition of SOX2 and/or neddylation may influence UC growth.

Arsenite (As^3+^) and arsenate (As^5+^) are the main forms of inorganic arsenic found in the environment and arsenite is considered to be the more toxic form [21]. Previously, our laboratory has developed an in-vitro cell culture model of arsenic-induced bladder cancer by exposing the immortalized non-tumorigenic cell line UROtsa to arsenite [22]. The transformed cell line express basal genes such as *KRT1*, *5*, *6*, *14* and *16*, and produce tumors in athymic mice which resemble UCs with areas of squamous differentiation [23,24,25]. The basal/squamous bladder cancers have a poor prognosis which could be due to increase expression of stemness related genes and an increase in resistance to chemotherapy. Recently, Wang et al. have shown that CIS-based chemotherapy can induce semi-squamatization, a form of squamous differentiation that does not achieve the level of terminal differentiation, in mice and human muscle invasive bladder cancers through lineage plasticity and this is associated with acquired chemoresistance [26]. The chemoresistant tumors express high levels of cathepsin H and its inhibition induces terminal differentiation and decreased growth of CIS resistant tumors.

In this study we determined if a combined treatment of PVD with CIS would influence the growth of As^3+^-transformed UROtsa cells and whether this occurs by inducing terminal differentiation in these cells.

## 2. Results

### 2.1. Proteomic Analysis of Parent and As^3+^-Transformed UROtsa Cells

We have previously transformed human UROtsa into tumorigenic cells using chronic exposures to 1 µM arsenite (As^3+^) and we found that As^3+^-transformed cells grow significantly more spheres or have more cancer initiating cells compared to the non-transformed parent UROtsa cells [22,27,28]. In the current study, we wanted to expand on previous results and identify proteins and pathways dysregulated during the As^3+^-transformation of UROtsa cells, particularly those that may enhance stemness. Urotsa cells were exposed to 1 µM As^3+^ over 40 passages along with control cells that were not exposed to As^3+^. At passages 5 (P5), 10 (P10), 20 (P20), and 40 (P40) cell pellets were collected for quantitative proteomic analysis. At P20 and every 5 passages after, cells were screened using soft-agar for anchorage-independent growth as an indicator of transformed cells. At P40, As_I, As_II, and As_III were forming colonies in soft auger, and tumor forming ability was confirmed in athymic nude mice. Appendix A display proteomic pathway analysis, PCA, loadings plots, and heatmaps for P5, P10, and P20 data compared to parent. Appendix A contains the original proteomics data. In total, 6262 proteins were quantified. The number of differentially expressed proteins (DEPs, >1.5 fold change and *p* < 0.05) in P5 vs. parent was 402, in P10 vs. parent was 384, in P20 vs. parent was 407, and in P40 vs. parent was 931. In the fully transformed cells (P40), proteins associated with organic substance biosynthesis, chromatin organization, chromatin condensation, mitochondrial gene expression and cell response to stress were elevated while proteins associated with organonitrogen compound metabolism, antigen processing and presentation, regulation of type I interferon production, and actin cytoskeleton organization were decreased (Figure 1A,B). PCA analysis shows that the parent (Par_I, Par_II, Par_III) and As^3+^-transformed (As_I, As_II, As_III) cells cluster separately with a majority of the variation in component 1 (74.1%) and component 2 displayed less (3.9%) variability between the parent and As^3+^-transformed cells (Figure 1C). The proteins most responsible for driving the variability between the parent and As^3+^-transformed cells are displayed in the corresponding loadings plots for component 1 and component 2 (Figure 1D,F) and Figure 1E shows a heatmap displaying the top 50 DEPs.

### 2.2. SOX2 Is Upregulated during As^3+^-Transformation of UROtsa Cells

Proteomics analysis revealed that levels of SOX2, a well-known stem marker, was elevated early (P5) in As^3+^-transformation and remained elevated into the fully transformed UROtsa cells (Figure 2A). We then measured the gene and protein levels in the fully transformed isolates (As_I, As_II, and As_III) compared to the parent (Figure 2B–D) using qPCR and Western blot, respectively. These results confirmed that both the gene and protein levels of SOX2 were elevated in the As^3+^-transformed cells compared to the parent UROtsa cells.

### 2.3. Morphology, Viability, and Gene Expression after Pevonedistat Treatment to UROtsa As_I Cells

We wanted to test if inhibiting SOX2 within the As^3+^-transformed cells would alter the morphology, reduce growth, or effect gene expression. To accomplish this, a dose-response was performed on the As_I cells by utilizing a potent SOX2 inhibitor called Pevonedistat (PVD) at 125 nM, 250 nM, and 500 nM concentrations for a duration of 24, 48, and 72 h. Figure 3A–L show brightfield photos of As_I cells following PVD treatments. At 24 h, even the lowest dose of PVD (125 nM) induced a morphological change in the cells compared to the DMSO control. The cells became quite large and squamous-like in appearance. As the treatments continued into 48 h, the cells remained large and developed white “cloudy” areas (black arrows) within the cultures that we believe may be cornified areas, particularly in the 250 nM and 500 nM concentrations. By 72 h, even the lowest concentration of PVD induced the white “cloudy” areas. Also, of note was that the highest dose never developed an intact monolayer of cells whereas the 125 nM and 250 nM doses had a relatively intact monolayers by 72 h. Higher magnification (40×) images of the As_I cells after PVD treatments is provided in Appendix A. The results of the crystal violet viability stain after 72 h PVD treatment shows that all three doses of PVD reduced the number of viable cells (Figure 3M,N). We next wanted to confirm that PVD inhibited expression of SOX2 and measured additional genes involved with squamous/terminal differentiation. Figure 4A demonstrates that PVD treatment significantly reduces (85–96%) the expression of SOX2 at all three doses and at each of the three time points. KLF4 is induced from PVD (Figure 4B) and is a gene involved in stemness, cell growth, and differentiation [29]. PVD also induced KRT6A, DSC2, IVL, SPRR3, and GRHL1 (Figure 4C–G) which are all associated with squamous/terminal differentiation [24,30,31,32].

To clarify if PVD treatment was inducing terminal differentiation throughout the entire culture (irreversible) a recovery experiment was set up utilizing the same doses and a duration of 72 h (Appendix A). After 72 h of treatment with PVD at the given doses, the cells were passaged (1:2) either in the presence (PVD) or absence (recovery) of the drug. One day after passage, the majority (~80–95%) of the cells had died when passaged directly into PVD while those passaged into regular media (recovery) appeared to have better viability (~50%, Appendix A). The cells were fed different concentrations of PVD or regular media (recovery) on day 3, then on day 5 images were taken again. Appendix A, demonstrates that almost all cells that were passaged into 250 and 500 nM PVD did not survive while those cells passaged into 125 nM PVD appeared to have some very large and irregular shaped living cells remaining. However, those split into regular media (recovery) appeared quite viable although the cells still appeared larger and had some areas of white “cloudiness” compared to the DMSO control (Appendix A). Noteworthy, the recovery cells had to be passaged at least 3 more times (4 total passages) for them to return to regular morphology and the cells passaged into 125 nM PVD did not survive a 2nd passage. These results suggest that PVD alone at the given doses for 72 h was not sufficient to terminally differentiate the entire culture of As^3+^-transformed UROtsa cells. These preliminary findings demonstrated that PVD at the initial higher doses were quite effective at reducing cell viability and SOX2 expression while also inducing terminal differentiation markers but we wanted to titer the drug down even further and also test the effects on the non-transformed parent. Appendix A demonstrates that PVD down to 50 nM (for 72 h) still induced the unique morphological changes seen with higher doses in the As_I cells. However, similar morphology within the parent could not be seen even up to 200 nM PVD. Based on these results, use of 50 nM PVD was the decided dose to be used in the remainder of the study.

### 2.4. Sphere Formation after PVD Treatment to UROtsa Parent and As_I Cells

Since SOX2 is linked stemness and sphere forming abilities [19], we wanted to assess sphere formation between the parent and As_I cells after PVD (50 nM) treatment. Figure 5A,C,E demonstrates that the DMSO-treated As_I cells formed over twice as many spheres as the DMSO-treated parent UROtsa cells, 176 vs. 83 spheres, respectively. However, when cells were treated with PVD (50 nM) the sphere forming ability of the As_I cells was attenuated back to parent levels (Figure 5A,D,E). When parent cells were treated with PVD, no change was observed in the number of spheres formed. It is clear from the images that the PVD-treated cells (Par and As_I) accumulated more single cells and/or debris. The single cells/debris is likely the result of cells sloughing off the spheres that did form. Taken together, these results suggest that PVD treatment reduces stemness of the As_I cells.

### 2.5. Morphology, Proliferation, and Apoptosis after PVD and Cisplatin Treatments to UROtsa Parent and As_I Cells

Cisplatin (CIS)- based treatments are commonly used to treat MIUC. Cisplatin has been shown to induce semi-squamatization, a level of squamous differentiation that does not reach terminal differentiation and is associated with acquired chemoresistance in MIUC [26]. We wanted to compare the effects of each treatment (PVD and CIS) either alone or in combination on the parent and As_I cells. Through preliminary experiments we found that 1 µM CIS for 72 h reduces viability (crystal violet) of the As_I cells to around 50% (Appendix A). Figure 6 shows the brightfield morphology photos taken after 72 h PVD, CIS, or PVD + CIS treatments to the parent and As_I UROtsa cells. In the parent, treatment with PVD had little effect to the morphology but CIS and PVD + CIS treatment made the cells somewhat enlarged and the monolayer was still relatively intact (Figure 6A,C,E,G). As previously shown, PVD treatment to the As_I cells resulted in large squamous looking cells with white “cloudy” areas (Figure 6D). The cells of the As_I became quite enlarged and somewhat squamous looking from CIS treatment and there were small holes in the monolayer (Figure 6F). However, the combination treatment of PVD + CIS to the As_I cells resulted in large squamous looking cells and the monolayer was severely disrupted with dense “cloudy” areas of cells (Figure 6H). Higher magnification images (20×) of the As_I cells treated with PVD and CIS are provided in Appendix A.

The As_I cells were more sensitive to the solo treatments of PVD or CIS compared to the parent cells when looking at proliferation rates but when the combination treatment was used the proliferation was reduced to similar levels between the parent and As_I cells (Figure 7A). We also performed an apoptosis assay to compare induction of cell death from PVD and CIS within the parent and As_I cells. Figure 7B,F–M shows that PVD, CIS, and PVD + CIS reduced the number of live cells in the As_I cells to a greater extent compared to the parent cells. Likewise, the drug treatments resulted in more dead cells in the As_I cells compared to the parent cells and a synergistic induction of dead cells was observed in the As_I cells from PVD + CIS treatment (Figure 7C,F,M). Early apoptosis was significantly elevated only from PVD treatment and late apoptosis was elevated from each treatment in the As_I cells comparehd to the parent cells (Figure 7M). The percentage of cells in late apoptosis was highest in CIS-treated (16.3%), then PVD + CIS-treated (4.9%), and PVD-treated (2.3%) As_I cells. These results demonstrate that the As_I cells are more sensitive to PVD, CIS, and PVD + CIS treatments in comparison to the non-transformed parent and that a combination of PVD + CIS results in synergistic elevation of cell death which suggests that PVD is enhancing CIS sensitivity within the As^3+^-transformed UROtsa cells.

### 2.6. Gene and Protein Expression in UROtsa Parent and As_I Cells after PVD and CIS Treatments

Since we had already observed that PVD treatment alone can induce morphological and molecular changes indicative of squamous/terminal differentiation. We wanted to assess whether the combination treatment of PVD + CIS would have an additive effect in inducing genes/proteins involved in terminal differentiation. The levels of SOX2 (gene and protein) were decreased from PVD, CIS, and PVD + CIS treatments in the parent cells (Figure 8A,C,H). In the As_I cells, SOX2 (gene and protein) was decreased from PVD and PVD + CIS treatments (Figure 8B,M,R). The protein expression of involucrin (IVL), a common terminal differentiation marker and component of the cornified envelope, was decreased from PVD, increased slightly (1.2-fold) from CIS, and unchanged from PVD + CIS treatment in the parent cells (Figure 8A,I). In the As_I cells, IVL protein was induced 2.9-fold from PVD, 4.6-fold from CIS, and 15.5-fold from PVD + CIS treatments (Figure 8B,S). The protein level of GRHL1, a transcription factor with roles in terminal differentiation, was slightly increased (1.3-fold) from CIS and PVD + CIS treatments in the parent cells (Figure 8A,J). In the As_I cells, GRHL1 was induced 1.3-fold from PVD and CIS treatments and induced 2.4-fold from PVD + CIS treatment (Figure 8B,T). The protein level of keratin 16 (KRT16), a protein associated with squamous differentiation and component of the cornified envelope, was not altered from PVD or CIS treatments in either the parent or As_I cells but was elevated 2.5-fold from PVD + CIS in the As_I cells (Figure 8A,B,K,U). The protein expression of annexin A1 (ANXA1), a protein involved in cell differentiation and also a component of the cornified envelope, was slightly induced (1.2-fold) from CIS treatment in the parent cells (Figure 8A,L). In the As_I cells, ANXA1 protein was elevated 1.2-fold from PVD and CIS treatments and increased up to 2.5-fold from PVD + CIS treatment (Figure 8B,V). Appendix A displays additional gene expression data in the As_I cells showing high induction of terminal differentiation markers from combination treatment of PVD + CIS but these genes were not elevated to the same amplitude in the parent cells (Appendix A). These findings suggest that treatments with PVD are decreasing stemness (SOX2) while inducing squamous/terminal differentiation markers. It also appears that the combination treatment (PVD + CIS) has a synergistic effect in inducing these squamous/terminal markers.

### 2.7. Gene and Protein Expression in Urospheres Derived from UROtsa As_I Cells

The hallmark characteristics of stem cells is their ability to self-renew and differentiate. We have previously published microarray results indicating that squamous/terminal differentiation markers are enriched in urospheres derived from As^3+^-transformed UROtsa cells and we believe that the stem cells may be responsible for regulating squamous differentiation in MIUC [28]. A majority of these squamous/terminal markers are highly inducible from PVD + CIS treatments indicating our ability to pharmacologically manipulate the stemness of As_I cells. The next step was to expand on and confirm these preliminary microarray findings by measuring the gene and protein levels of some stem-like markers and squamous/terminal differentiation markers in urospheres derived from UROtsa As_I cells. Figure 9A,G demonstrates that SOX2 protein was downregulated in As_I urospheres compared to the As_I monolayer cultures. Meanwhile, the proteins involved with squamous/terminal differentiation (IVL, GRHL1, KRT16, and ANXA1) were highly enriched in the urospheres (Figure 9A,H–K). Additional gene expression results are presented in Appendix A. These data show some stem genes either increase (*KLF4*, *CD24*, *ALDH1A1*) and some decrease (*OCT4*, *CD44*, *ALDH3A1*) and all the terminal differentiation/cornification genes analyzed were all elevated (*SPRR1A*, *SPRR2A*, *SPRR3*, *GRHL3*, *CERS3*) in the As_I urospheres compared to monolayer As_I cells. Similar to pharmacological inhibition of SOX2, these data show that when SOX2 is downregulated in the spheres, the squamous/terminal differentiation markers are elevated.

### 2.8. Immunohistochemical Analysis of Tumor Heterotransplants Derived from UROtsa As_I Cells

We performed immunohistochemical (IHC) analysis within tumor heterotransplants derived from UROtsa As_I cells to get a better understanding of protein localization in vivo for SOX2, IVL, and GRHL1. SOX2 protein was found in the nuclei in both well-differentiated (squamous) areas and non-differentiated (basal) areas of the tumors (Figure 10A). IVL protein was mainly found in the squamous areas of the tumors and was cytoplasmic (Figure 10B). GRHL1 protein was nuclear and was mainly localized to the squamous areas of the tumors (Figure 10C). Staining intensities and percentage of cells staining for each protein is listed in Table 1.

## 3. Discussion

Cancer stem cells (CSCs) are a subpopulation of tumor cells that self-renew themselves and differentiate into the cell types present in the tumor [33]. They play a role in the tumor development, progression, metastasis, recurrence and resistance to chemotherapy and radiotherapy [13,34,35,36]. Identification of CSCs can be problematic as the expression of CSC markers is context dependent and varies among different cancers and within a cancer type, thus indicating a high level of heterogeneity. The expression OCT4, KLF4, c-MYC, and SOX2 are routinely overexpressed in CSCs and it is thought that these transcription factors drive development, proliferation, and stemness of tumors [37].

In UCs, muscle invasive and non-muscle invasive cancers arise from distinct progenitor cells and have different expression profiles of stem cell markers [38]. In a study performed by Teixeira et al., a two gene stem-like signature (*SOX2* and *ALDH2*) was identified that allowed distinguishing between NMIUC and MIUCs [39]. In our proteomic analysis of the As^3+^-transformed UROtsa cells, SOX2 was identified as a protein that was highly expressed in the transformed cells compared to the UROtsa parent cell. The expression level of SOX2 increased early during the transformation process and showed the highest expression level once the cells were transformed as determined by soft agar assay. This suggests that *SOX2* may be one of the key genes involved in the development of the basal subtype of UC which is represented by our As^3+^-transformed model system. In another study, transformation of the SV40 immortalized urothelial cells HUV1 with arsenite resulted in an increase in expression of SOX2 [40]. Furthermore, this study also demonstrated that the level of SOX2 was slightly elevated in the urine samples from individuals exposed to arsenite in Bangladesh and in the urine samples of individuals diagnosed with urothelial bladder cancer. In another study, Migata et.al show a relationship between epithelial-mesenchymal transition (EMT) and cancer stemness and a decrease in E-cadherin increased the expression of SOX2 and NANOG in bladder cancer cell lines [41]. In addition, the study also demonstrated an increased expression of SOX2 in MIUC with a decrease in the expression of E-cadherin. Thus, it seems that an increased expression of SOX2 promotes the development of an aggressive form of bladder cancer with more mesenchymal features.

Previous studies have shown that treatment of cancers with PVD, an inhibitor of SOX2 can affect the growth and drug resistance of the cancer cells [19]. In our study we found that a decrease in the expression of SOX2 with PVD treatment in the As_I cells resulted in the differentiation of the cells. Gene expression analysis of the treated cells showed an increase in expression of genes associated with squamous/terminal differentiation such as *KRT6*, *KRT16*, *KRT14*, *IVL*, *DSC2*, *GRHL1* and *2*, *SPRR1A*, *2A* and *3*. The role of KRTs in the differentiation of epithelial cells is well established and in our previous studies, we showed that transformation of the UROtsa cells with arsenite induces the expression of basal KRTs 5, 6, 14 and 16 and the expression is associated with areas of squamous differentiation in tumor hetero-transplants [24,25,28,42,43]. Other markers such as IVL and DSC2 are also expressed in UCs with squamous differentiation [26,30,44,45]. Furthermore, DSC2 has been shown to be a highly specific and sensitive marker that distinguishes UC with squamous differentiation from pure UC [30]. The role of GRHL family of transcription factors in the development of UC with squamous/terminal differentiation is not clear. These factors can either promote the development of cancer (oncogenic) or they can serve as tumor suppressors depending upon which GRHL isoform is expressed and the type of cancer. In UCs, all three GRHL isoforms are frequently mutated [46]. These transcription factors control morphogenesis and epithelial differentiation and it is possible that their expression may contribute to squamous/terminal differentiation in UC. In fact, it has been demonstrated that there is an elevated expression of GRHL3 in well-differentiated UC cell lines compared to those that are less differentiated [47]. The small proline-rich proteins (SPRR) family are markers of terminal differentiation in the skin and are found in the cornified envelope. These genes have been found to be expressed in the bladder and a recent study found these genes expressed in areas of squamous differentiation in UCs [48,49]. The Warrick et.al study used bulk RNA-Seq analysis of macro-dissected regions of pure UC vs. squamous differentiated regions from mixed histology UC. They demonstrated an enrichment of genes associated with epidermal development (*GRHL1*, *CERS3*, *ANXA1*, *KRT14*, *IVL*, *FLG*, *SPRR1A*, *SPRR2A*, *SPRR3*), epithelial cell differentiation (*SPRR1A*, *SPRR2A*, *SPRR3*, *KRT14*, *KRT16*, *IVL*, *FLG*, *ANXA1*, *CERS3*), and formation of the cornified envelope (*ANXA1*, *SPRR1A*, *SPRR2A*, *SPRR3*, *IVL*) in the squamous areas [49]. These data agree well with our in vitro results and indicates that when we pharmacologically inhibit SOX2 using PVD or PVD + CIS, there is induction of genes associated with squamous/terminal differentiation.

The expression of SOX2 is also required to maintain the stemness of cells which can be measured by the sphere forming assay. In our study the expression of SOX2 was decreased in the spheres formed by the As_I cells when compared to the monolayer and this decrease was accompanied by an increase in the expression of genes involved in squamous/terminal differentiation. Treatment of the As_I cells and the UROtsa parent cells with PVD decreased the ability of the As_I cells to form spheres without any effect on the parental cells. This suggests that SOX2 is probably required for the initial proliferation and sphere formation but upon development of the spheres, the expression is no longer required, and it decreases resulting in the terminal differentiation of the cells within the spheres.

The state of differentiation of cancers is important not only for its histo-pathological classification but also for the determination of the cancer’s aggressiveness and behavior. Differentiation therapy for solid tumors was first proposed as a therapeutic option by G. B. Pierce who suggested that malignant cells could differentiate into non-malignant cells [50,51]. This form of therapy may not necessarily result in the eradication of all cancer cells by converting them into normal cells, but it may convert the cancer to a lower grade which could ultimately improve the prognosis of an individual. Cisplatin-based chemotherapy is the first-line treatment for patients suffering from MIUC [52]. Recently, Wang et al. have demonstrated that treatment of MIUC with CIS results in the increase in semi-squamatization, a form of squamous differentiation that doesn’t reach the level of terminal differentiation [26]. The authors demonstrate that semi-squamatization leads to chemoresistance and they propose that semi-squamatization may increase susceptibility of the cancer to differentiation therapy. Since PVD in our study increases the expression of genes associated with squamous/terminal differentiation, we performed a study with a combination of CIS and PVD to determine the effect of the treatment on squamous/terminal differentiation. The combined treatment decreased the viability of the As_I cells when compared to the individual treatments and increased squamous features. Gene and protein expression analysis showed an increase in expression of genes involved in squamous/terminal differentiation when compare to the individual treatments suggesting a synergistic effect of the two drugs. Pevonedistat has shown activity against various solid cancer lines, human tumor xenografts as well as hematological malignancies [53]. A combination of PVD with CIS has shown that PVD can enhance the anti-tumor effect of CIS in UC cell lines by increasing DNA damage and potentiate CIS-induced apoptosis [54]. Resistance to chemotherapeutic drugs is a major clinical problem and the basal/squamous gene expression subtype of MIUC seem to more resistant to platinum-based therapies. Thus, based on our and other studies, it seems that treatment with both PVD and CIS can potentially increase the sensitivity of the tumor to chemotherapeutic drugs.

Differentiation therapy is an area in research that relies on terminally differentiating cancer cells until they are unable to divide and lose tumorigenicity. Many anticancer drugs target rapidly dividing cells and may partially explain why squamous differentiation in tumors can be associated with chemo-resistance. Nonetheless, it would be highly beneficial to induce terminal differentiation throughout the entire UC tumor to impede growth and metastatic ability. Future in vivo animal studies will address if PVD with CIS can attenuate tumor growth and metastatic potential. Our data suggests that the combined treatment of PVD with CIS can induce squamous/terminal differentiation in a model of the basal MIUC. This combined treatment may reduce the proliferation of the cells and prevent metastasis, thereby improving the patient prognosis of MIUC.

## 4. Materials and Methods

### 4.1. Animals

This study adhered to all recommendations dictated in the Guide for the Care and Use of Laboratory Animals of the NIH. The protocol was approved by The University of North Dakota Animal care Committee (IACUC# 1911-1C). Athymic nude (NCR-nu/nu) female mice were used in these studies. The mice were housed five to a cage at 22 °C under a 12-h light/dark cycle. Food and water was available *ad libitum*. The As^3+^-transformed UROtsa cells were injected subcutaneously (SQ) into 5 nude (NCr-nu/nu) mice per group. The SQ injection of transformed cells has been described previously in detail [24,55]. The tumor size was assessed weekly using a ruler and the animals were sacrificed when the size of the tumor was approximately 1.5–1.8 cm or when dictated by clinical conditions. The animals were euthanized by CO_2_ asphyxiation and conformed to American Veterinary Medical Association Guideline on Euthanasia. Death was confirmed by ascertaining cardiac and respiratory arrest following which the tumors were harvested. Care was taken to ensure that there was no distress to the animals during the procedure.

### 4.2. Cell Culture

The UROtsa parent cells and three of the As^3+^-transformed isolates (As_I, As_II, and As_III) were cultured in in Dulbeco’s modified Eagle’s medium (DMEM) supplemented with 5% *v/v* fetal bovine serum (FBS) as described previously [22]. The cells were sub-cultured at a 1:10 ratio using trypsin-EDTA and the cultures were fed fresh growth medium every three days. The UROtsa parent cell line has been authenticated using short tandem repeat (STR) analysis [56]. The As^3+^-transformed isolates used in the current study have been previously characterized for their ability to form colonies in soft agar, form spheroids when grown in ultra-low attachment flasks and form tumors when injected subcutaneously in immune-compromised mice [22,24,27,43,56]. For drug treatments, UROtsa parent and As_I were grown to ~70% confluence. The cells were then exposed to either dimethyl sulfoxide (DMSO, drug vehicle), Pevonedistat (PVD), cisplatin (CIS), or a combination of PVD and CIS (PVD + CIS) for the indicated durations. The concentrations of the drugs were chosen based on preliminary studies.

### 4.3. Quantitative Proteomics

Total protein from cell pellets was reduced, alkylated, and purified by chloroform/methanol extraction prior to digestion with sequencing grade modified porcine trypsin (Promega). Tryptic peptides were labeled using tandem mass tag isobaric labeling reagents (Thermo) following the manufacturer’s instructions and combined into one 16-plex TMTpro sample group. The labeled peptide multiplex was separated into 46 fractions on a 100 × 1.0 mm Acquity BEH C18 column (Waters) using an UltiMate 3000 UHPLC system (Thermo, Waltham, MA, USA) with a 50 min gradient from 99:1 to 60:40 buffer A:B ratio under basic pH conditions, and then consolidated into 18 super-fractions. Each super-fraction was then further separated by reverse phase XSelect CSH C18 2.5 um resin (Waters Corporation, Milford, MA, USA) on an in-line 150 × 0.075 mm column using an UltiMate 3000 RSLCnano system (Thermo). Peptides were eluted using a 75 min gradient from 98:2 to 60:40 buffer A:B ratio. Eluted peptides were ionized by electrospray (2.4 kV) followed by mass spectrometric analysis on an Orbitrap Eclipse Tribrid mass spectrometer (Thermo) using multi-notch MS3 parameters. MS data were acquired using the FTMS analyzer in top-speed profile mode at a resolution of 120,000 over a range of 375 to 1500 *m*/*z*. Following CID activation with normalized collision energy of 35.0, MS/MS data were acquired using the ion trap analyzer in centroid mode and normal mass range. Using synchronous precursor selection, up to 10 MS/MS precursors were selected for HCD activation with normalized collision energy of 65.0, followed by acquisition of MS3 reporter ion data using the FTMS analyzer in profile mode at a resolution of 50,000 over a range of 100–500 *m*/*z*. Buffer A = 0.1% formic acid, 0.5% acetonitrile. Buffer B = 0.1% formic acid, 99.9% acetonitrile. Both buffers adjusted to pH 10 with ammonium hydroxide for offline separation.

### 4.4. Bioinformatic Analysis of Proteomic Data

Proteins were identified and MS3 reporter ions quantified using MaxQuant (Max Planck Institute, version 1.6.12.0) against the UniprotKB Homo sapiens (March 2021) database with a parent ion tolerance of 3 ppm, a fragment ion tolerance of 0.5 Da, and a reporter ion tolerance of 0.003 Da. Scaffold Q+S (Proteome Software, version 5.2.2) was used to verify MS/MS based peptide and protein identifications. Protein identifications were accepted if they could be established with less than 1.0% false discovery and contained at least 2 identified peptides. Protein probabilities were assigned by the Protein Prophet algorithm [57]. Protein TMT MS3 reporter ion intensity values are assessed for quality and normalized using proteiNorm [58]. The data was normalized using cyclic loess and statistical analysis was performed using Linear Models for Microarray Data (limma) with empirical Bayes (eBayes) smoothing to the standard errors [59]. Proteins with an FDR adjusted *p*-value < 0.05 and a fold change >1.5 were considered to be significant. Principle component analysis (PCA) and heatmaps were generated using MetaboAnalyst 5.0 software [60]. Data were normalized using pareto scaling and compared by T-test with an applied false discovery rate (FDR). Heatmap analysis with data clustering was performed using Euclidean distance measuring and a Ward clustering algorithm. Pathway analysis was generated using g:Profiler using differentially expressed proteins (increased or decreased) at each passage as the input [61]. For pathway analysis and heatmap generation, a fold change greater than 1.5-fold and an FDR adjusted *p*-value < 0.05 was used.

### 4.5. Crystal Violet Staining for Cell Viability

To assess cell viability, cells in a 12-well plate were stained with 500 µL of 0.5% crystal violet (Sigma-Aldrich Corp., St. Louis, MO, USA) in 20% methanol for 15 min at room temperature. The plate was washed five times in fresh water and allowed to dry. The cells were lysed in a 0.1 M sodium citrate in 25% ethanol solution (pH 4.2) for 15 min on a shaker. The absorbance of the solution was measured using microplate spectrophotometer at a wavelength of 595 nm.

### 4.6. Tag-It Violet Proliferation Assay

Six million cells were incubated with 1 mL Phosphate Buffered Saline (PBS) that contained 5 µM Tag-IT violet dye (BioLegend, #425101) for 30 min at 37 °C, after which five milliliters of DMEM medium with 5% FBS was added to terminate the reaction, and the cells were centrifuged at 2000 rpm for 5 min. Cells were resuspended in fresh DMEM medium with 5% FBS and seeded into six-well plates at a density of 300,000 cells per well. Forty-eight hours afterwards, the cells were treated with either DMSO, PVD, CIS, or PVD with CIS. Seventy-two hours after treatment, the cells were trypsinized and collected in Falcon tubes (Corning, #352058, Glendale, AZ, USA) in 500 µL fluorescence-activated cell sorting (FACS) buffer (PBS + 5% FBS). Samples were acquired on a Sony (SH800) flow cytometer and analyzed with FlowJo software (version v10.9.0).

### 4.7. Sphere Formation and Quantitation

UROtsa Parent and As_I cells were trypsinized and washed three times with PBS. After washing, cells were taken up in 1 mL of a serum free medium consisting of a 1:1 mixture of Dulbecco’s modified Eagle’s medium (DMEM) and Ham’s F-12 growth medium supplemented with selenium (5 ng/mL), insulin (5 μg/mL), transferrin (5 μg/mL), hydrocortisone (36 ng/mL), triiodothyronine (4 pg/mL), and epidermal growth factor (10 ng/mL) that has previously been described [28,62,63]. Cells were then run through a 70-micron cell strainer to get a homogenous single cell population. The cells were then counted by trypan blue and were seeded 25,000 cells per T-25 flask (ultra-low attachment) in 5 mL of the serum-free media containing DMSO or PVD (50 nM). After 3 days, spheres were fed 2 mL fresh media containing either DMSO or 50 nM PVD. The spheres were allowed to grow for a total of 6 days before harvesting for counting or RNA and protein isolation. For RNA and protein isolation, the spheres were pelleted by centrifugation and lysed in RLT plus (RNA) or RIPA (protein). To quantify spheres, the media containing spheres was collected from the T-25 flasks into a 15 mL conical tube. The T-25 flask was washed with 3 mL of serum free media and was also added to the 15 mL conical tube. Spheres were centrifuged at 150× *g* for 5 min. The supernatant was removed and 500 µL of serum-free media was added and the pipette was used to mix spheres. Using a fine-tip sharpie pen, the bottom of a 96-well plate was divided into four quadrants. 50 µL spheres/media was divided into 10 wells of the 96-well plate. The spheres within each quadrant of each of the 10 wells was counted under a microscope using 4× power objective. The sphere concentration was calculated by dividing the spheres counted per well by the volume (50 µL). The number of total spheres counted was calculated by multiplying the concentration of spheres by the total volume (500 µL).

### 4.8. Apoptosis Assay

The cells were cultured in media containing DMSO, PVD, CIS, or PVD + CIS, trypsinized, then doubly stained for annexin V and propidium iodide (PI) using a Pacific blue annexin V apoptosis kit (Biolegend, catalog no. 640928) according to the manufacturer’s instruction and analyzed by flow cytometry (Sony SH800). The data were analyzed with FlowJo software (version v10.9.0).

### 4.9. RNA Isolation and Real-Time PCR Analysis

The cells were washed twice with PBS then total RNA was isolated from cells using the RNeasy Plus Mini Kit, Qiashredders, and QiaCube instrument following the manufacturer’s protocols (Qiagen, Hilden, Germany). Total RNA (1 µg) was transcribed into cDNA using LunaScript^®^ RT SuperMix Kit (New England Biolabs #E3010L, Ipswich, MA, USA) according to the manufacturer’s protocol. cDNA (2 µL) was combined with 0.2 µM primers and Luna^®^ Universal qPCR Master Mix (New England Biolabs # M3003E) according to the manufacturer’s protocol in a total volume of 20 µL. qPCR gene amplification was monitored by SYBR Green fluorescence using BioRad CFX96 Touch Real-Time PCR Detection System. qPCR cycle conditions were 1 cycle of 2 min at 95 °C, and 40 cycles of 5 s at 95 °C and 30 s at the annealing temperature of 60 °C. Expression levels were determined from the threshold cycle (Ct) values using the method of 2-∆∆Ct using 18S as the reference control gene. Primers were obtained from Integrated DNA Technology (IDT, Coralville, IA, USA) and are listed in Appendix A.

### 4.10. Simple Western Blot Analysis

The use of Simple Western blot to measure protein expression has been previously described [64,65,66,67]. Briefly, the cells were washed twice with PBS then dissolved in 1X Radio-immunoassay Precipitation Assay (RIPA) lysis buffer supplemented with PMSF, protease inhibitor cocktail, and sodium orthovandate (Santa Cruz Biotechnology, Dallas, TX, USA). The cell suspension was sonicated and the lysate was centrifuged to remove cellular debris. Protein lysates were quantified using the Pierce BCA protein assay kit (Thermo-Scientific Pierce). Diluted protein lysates were combined with 5X fluorescent master mix (ProteinSimple, San Jose, CA, USA) which contains reducing agent (dithiothreitol), fluorescent standards, and a system control protein (26 kDa). Denatured (95 °C for 5 min) lysates were separated and immunodetection of target proteins was performed using a capillary-based Jess Simple Western instrument (ProteinSimple, San Jose, CA, USA) according to the manufacturer’s protocol. The ProteinSimple system control protein (26 kDa) served as an internal control and was also used to normalize protein expression. Duplicates of each sample (4 µL) were analyzed for target protein expression. The protein lysate concentrations used to detect each target protein and antibodies/dilutions are listed in Appendix A. The uncropped blots are provided in Appendix A.

### 4.11. Immunohistochemical Staining

Subcutaneous tumor tissue obtained from mice injected with As^3+^-transformed UROtsa cells was used in this study. Serial sections were cut at 3–5 µm and immersed in preheated Target Retrieval Solution (citrate pH 6, Agilent) in a steamer for 20 min. The sections were allowed to cool to room temperature and immersed into TBS-T for 5 min. The primary antibodies used in this study along with their dilutions and catalogue numbers are listed in Appendix A. The primary antibodies were localized using Dako peroxidase conjugated EnVision plus for rabbit or mouse primary antibodies at room temperature for 30 min. Liquid Diaminobenzidine (Dako) was used for visualization. Counter staining was performed using hematoxylin. Slides were rinsed in distilled water, dehydrated in graded ethanol, cleared in xylene, and cover slipped. Two pathologists judged the presence and degree of immune-reactivity in the specimens. The scale used was 0 to +3 with 0 indicating no staining, +1 staining of mild intensity, +2 staining of moderate intensity, and +3 staining of strong intensity.

### 4.12. Statistical Analysis

Unless otherwise stated, experiments were performed in either duplicate or triplicate and the results are expressed as the mean ± SEM. Statistical analyses were performed using GraphPad Prism^®^ software version 9.5.1 using *t*-test, one-way ANOVA, or two-way ANOVA. The level of significance was *p* < 0.05.

## Figures and Tables

**Figure 1 ijms-24-09149-f001:**
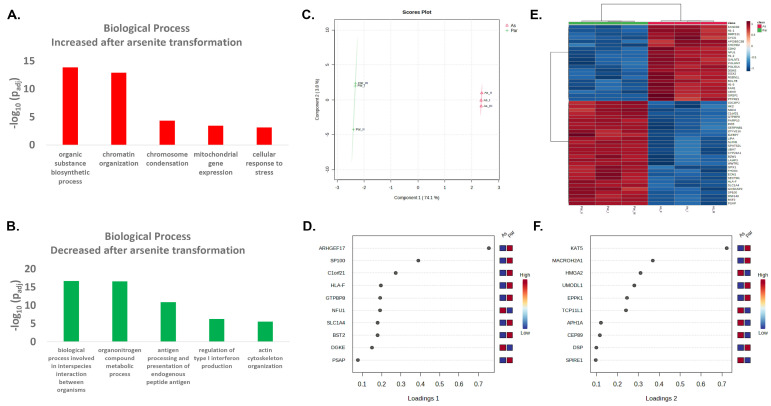
Proteomic analysis after As^3+^-transformation of UROtsa cells. (**A**) The top five biological process pathways upregulated after As^3+^-transformation of UROtsa cells at passage 40. (**B**) The top five biological process pathways downregulated after As^3+^-transformation of UROtsa cells (passage 40). (**C**) Principle component analysis (PCA) of proteomic data comparing variability between the parent (Par_I, Par_II, Par_III) and As^3+^-transformed (As_I, As_II, As_III) UROtsa cells at passage 40. (**D**) Loadings plot of proteomic data showing the main proteins responsible for driving the variability in component 1 between the parent and As^3+^-transformed cells at passage 40. (**E**) Heatmap displaying the top 50 differentially expressed proteins (DEPs) between the parent and As^3+^-transformed UROtsa cells having greater than 1.5-fold change and *p* < 0.05. (**F**) Loadings plot of proteomic data showing the main proteins responsible for driving the variability in component 2 between the parent and As^3+^-transformed cells at passage 40.

**Figure 2 ijms-24-09149-f002:**
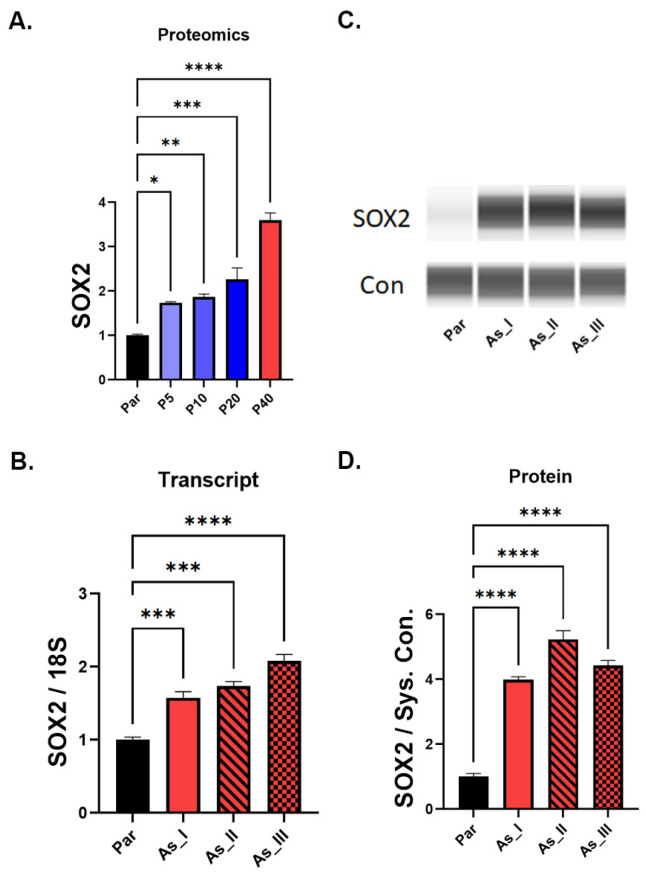
Elevated SOX2 in As^3+^-transformed UROtsa cells. (**A**) Resulting proteomic data for SOX2 expression between the UROtsa parent (Par) and various passages (P5, P10, P20, P40) in 1 µM As^3+^. (**B**) Transcript levels of *SOX2* gene in parent and fully As^3+^-transformed (P40) UROtsa cell isolates. (**C**) Representative Western blot image of SOX2 protein expression between the parent and fully As^3+^-transformed (P40) UROtsa cell isolates. (**D**) Quantification of Western blot data between the parent and fully As^3+^-transformed (P40) UROtsa cell isolates. All data is plotted as fold-change compared to the non-transformed parent. Gene expression was normalized to the 18S housekeeping gene and protein expression was normalized to the Jess system control. The measurements were performed in triplicates for gene data and duplicates for protein data. The values reported are mean ± SEM. An ordinary one-way ANOVA was performed followed by a Dunnett’s post-hoc test. Asterisks indicate significant differences from the parent (* *p* < 0.05, ** *p* ≤ 0.01, *** *p* ≤ 0.001, **** *p* ≤ 0.0001).

**Figure 3 ijms-24-09149-f003:**
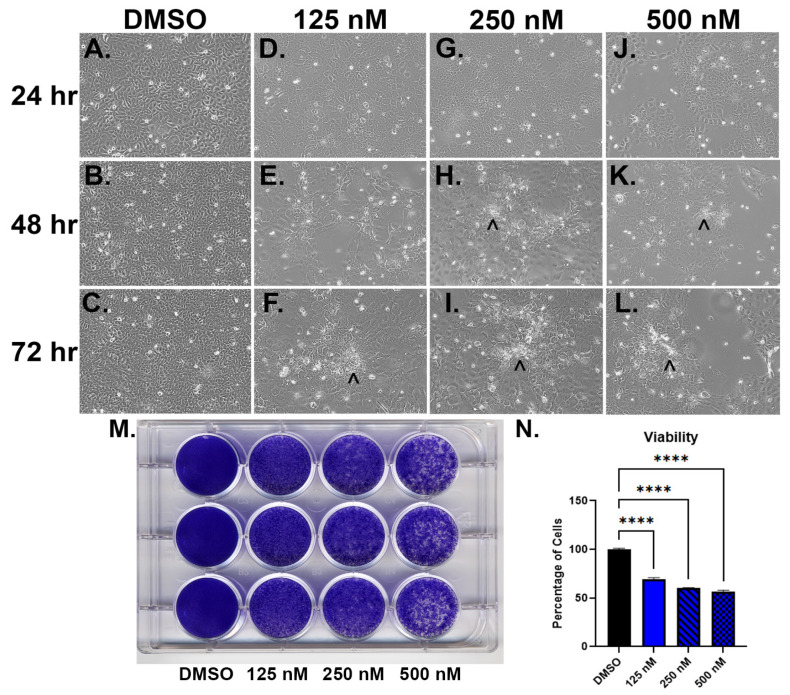
Morphology and viability of UROtsa As_I cells after PVD treatment. (**A**–**L**) Brightfield microscope images of As_I cells taken at 10× magnification. (**A**–**C**) DMSO control at 24, 48 and 72 h. (**D**–**F**) 125 nM PVD at 24, 48, and 72 h. (**G**–**I**) 250 nM PVD at 24, 48, and 72 h. (**J**–**L**) 500 nM PVD at 24, 48, and 72 h. (**M**) Image of crystal violet results after 72 h exposures to DMSO, 125 nM PVD, 250 nM PVD, and 500 nM PVD. (**N**) Crystal violet data plotted as percentage of viable cells compared to the DMSO control. Forty-eight hours prior to treatments, 100,000 cells were seeded per well of a 12-well plate and allowed to grow. The cells were then treated for the indicated times. Cells were seeded in triplicate wells for each treatment and the values reported are mean ± SEM. An ordinary one-way ANOVA was performed followed by a Dunnett’s post-hoc test. Asterisks indicate significant differences from the parent (**** *p* ≤ 0.0001). Black arrows (^) indicate white “cloudy” areas.

**Figure 4 ijms-24-09149-f004:**
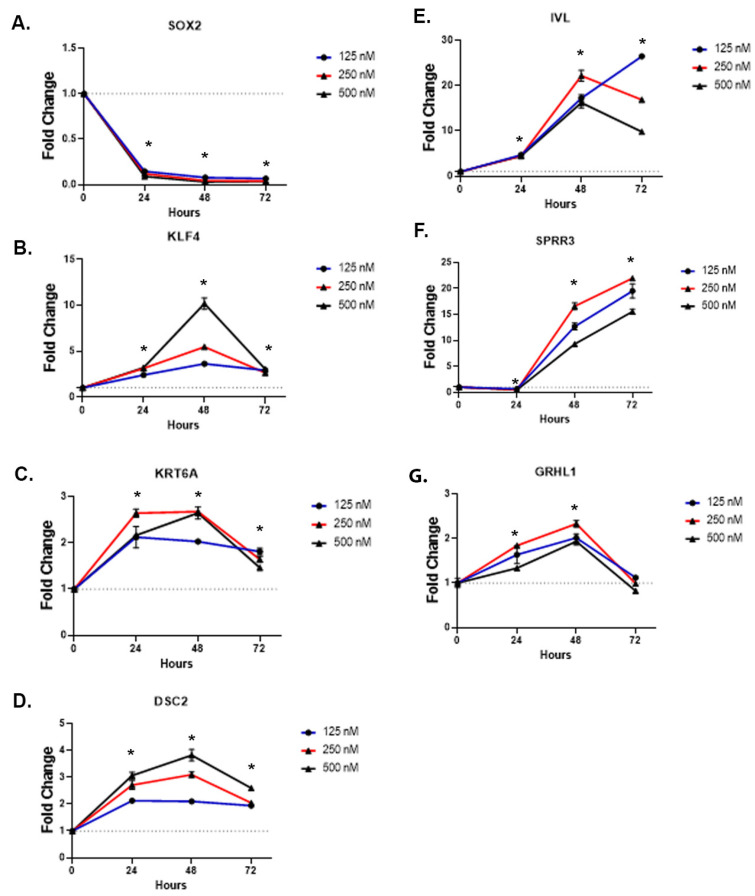
Gene expression in UROtsa As_I cells after time course PVD treatment. Cells were treated with either DMSO (dotted line), 125 nM PVD (blue bar), 250 nM PVD (red bar), or 500 nM PVD (black bar) for 24, 48 and 72 h. The cells were washed twice with PBS, mRNA was isolated, and qPCR was performed. Gene expression results for (**A**) *SOX2*, (**B**) *KLF4*, (**C**) *KRT6A*, (**D**) *DSC2*, (**E**) *IVL*, (**F**) *SPRR3*, (**G**) *GRHL1*. Forty-eight hours prior to treatments, 100,000 cells were seeded per well of a 12-well plate and allowed to grow. The cells were then treated for the indicated times. Data represents triplicate measurements that are reported as fold-change compared to the DMSO control for each point and values are mean ± SEM. Gene expression was normalized to the 18S housekeeping gene. An ordinary one-way ANOVA was performed followed by a Dunnett’s post-hoc test for each time point. Asterisks indicate significant differences from the DMSO control for each time point (* *p* < 0.05).

**Figure 5 ijms-24-09149-f005:**
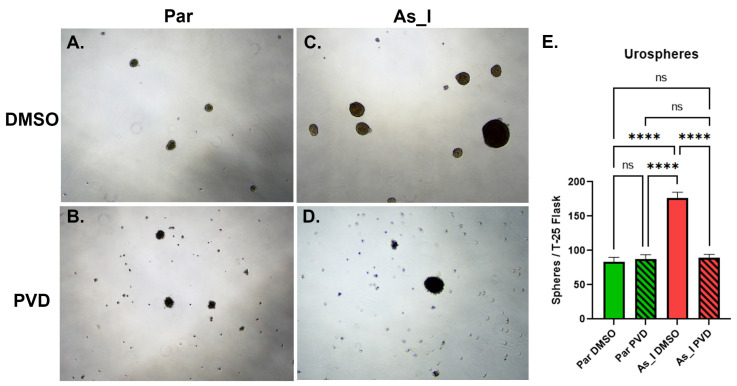
Urosphere formation after PVD treatment to UROtsa parent and As_I cells. Cells were seeded at 25,000 cells per T-25 ultra-low attachment flask in serum-free media containing DMSO or 50 nM PVD and allowed to grow for 6 days. Brightfield images using 4× magnification were taken for (**A**) Parent UROtsa in DMSO, (**B**) Parent UROtsa in PVD, (**C**) As_I UROtsa in DMSO, and (**D**) AS_I UROtsa in PVD. (**E**) Quantification of urospheres per T-25 flask. Triplicate measurements were performed for each treatment and the values represent mean ± SEM. An ordinary one-way ANOVA was performed followed by a Tukey’s post-hoc test. Asterisks indicate significant differences between the groups (**** *p* ≤ 0.0001) and “ns” indicates not significant.

**Figure 6 ijms-24-09149-f006:**
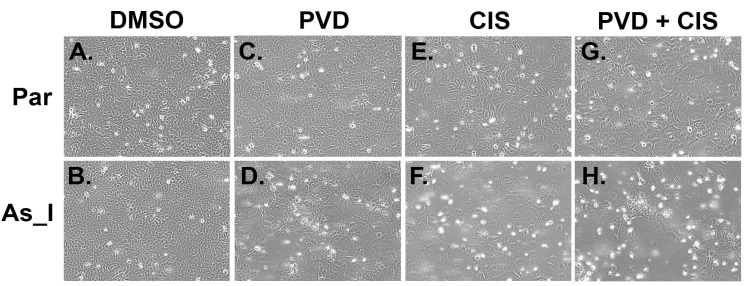
Morphology after 72-h PVD and CIS treatments to UROtsa parent and As_I cells. (**A**–**H**) Brightfield microscope images of parent and As_I cells taken at 10× magnification. (**A**) parent DMSO, (**B**) As_I DMSO, (**C**) parent 50 nM PVD, (**D**) As_I 50 nM PVD, (**E**) parent 1 µM CIS, (**F**) As_I 1 µM CIS, (**G**) parent 50 nM PVD + 1 µM CIS, (**H**) As_I 50 nM PVD + 1 µM CIS. Forty-eight hours prior to treatments, 300,000 cells were seeded per well of a 6-well plate and allowed to grow. The cells were then treated for 72 h.

**Figure 7 ijms-24-09149-f007:**
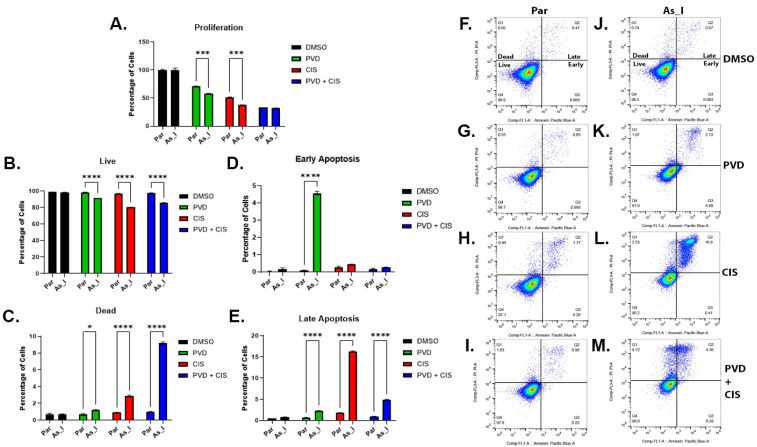
Proliferation and apoptosis after PVD and CIS treatments to UROtsa parent and As_I cells. (**A**) Flow cytometric analysis of cell proliferation using Tag-it Violet labeling of cells 72 h after DMSO (black bars), 50 nM PVD (green bars), 1 µM CIS (red bars), or 50 nM PVD + 1 µM CIS (blue bars) treatments in parent (Par) and As_I UROtsa cells. (**B**–**M**) Flow cytometric analysis of apoptosis 48 h after DMSO, 50 nM PVD, 1 uM CIS, or 50 nM PVD + 1 µM CIS treatments in parent and As_I UROtsa cells. (**B**) live cells, (**C**) dead cells, (**D**) early apoptosis, (**E**) late apoptosis. Flow cytometric figures generated from (**F**) parent DMSO, (**G**) parent PVD, (**H**) parent CIS, (**I**) parent PVD + CIS, (**J**) As_I DMSO, (**K**) As_I PVD, (**L**) As_I CIS, (**M**) As_I PVD + CIS treatments. Quadrant 1 (Q1) is percentage of dead cells. Quadrant 2 (Q2) is percentage of cells in late apoptosis. Quadrant 3 (Q3) is percentage of cells in early apoptosis. Quadrant 4 (Q4) is percentage live cells. Forty-eight hours prior to treatments, 300,000 cells were seeded per well of a 6-well plate and allowed to grow. Duplicate measurements were performed for each treatment and the data is reported as percentage of cells where the values represent mean ± SEM. A two-way ANOVA was performed followed by a Sidak’s multiple comparison test to analyze significant differences between the parent and As_I cells. Asterisks indicate significant differences between the two cells (* *p* < 0.05, *** *p* ≤ 0.001, **** *p* ≤ 0.0001).

**Figure 8 ijms-24-09149-f008:**
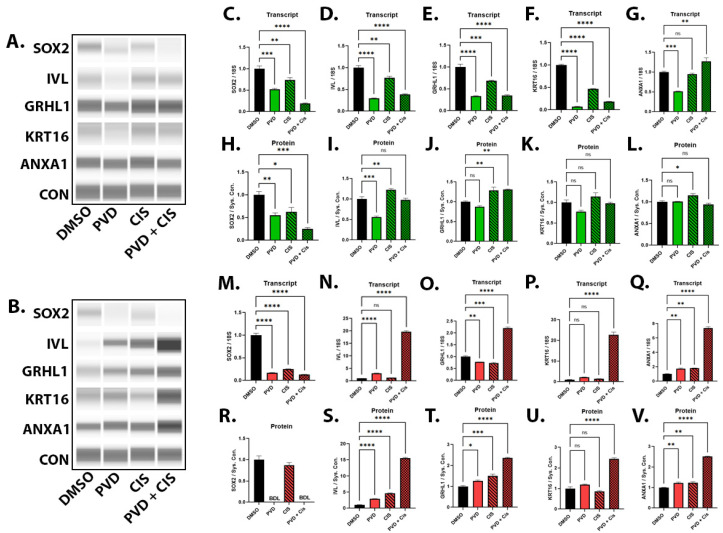
Gene and protein expression in UROtsa parent and As_I cells after PVD and CIS treatments. Cells were treated for 72 h with DMSO, 50 nM PVD, 1 µM CIS, or 50 nM PVD + 1 µM CIS treatments in parent (green bar graphs) and As_I UROtsa cells (red bar graphs). Western blot images for parent (**A**) and As_I (**B**) UROtsa cells. Parent transcript levels for (**C**) SOX2, (**D**) IVL, (**E**) GRHL1, (**F**) KRT16, (**G**) ANXA1. Parent protein quantification for (**H**) SOX2, (**I**) IVL, (**J**) GRHL1, (**K**) KRT16, (**L**) ANXA1. As_I transcript levels for (**M**) SOX2, (**N**) IVL, (**O**) GRHL1, (**P**) KRT16, (**Q**) ANXA1. As_I protein quantification for (**R**) SOX2, (**S**) IVL, (**T**) GRHL1, (**U**) KRT16, (**V**) ANXA1. All data is plotted as fold-change compared to the DMSO control. Gene expression was normalized to the 18S housekeeping gene and protein expression was normalized to the Jess system control. The gene measurements were performed in triplicates while protein measurements were performed in duplicates and the values reported are mean ± SEM. An ordinary one-way ANOVA was performed followed by a Dunnett’s post-hoc test. Asterisks indicate significant differences from the DMSO control (* *p* < 0.05, ** *p* ≤ 0.01, *** *p* ≤ 0.001, **** *p* ≤ 0.0001). BDL, below detection levels.

**Figure 9 ijms-24-09149-f009:**
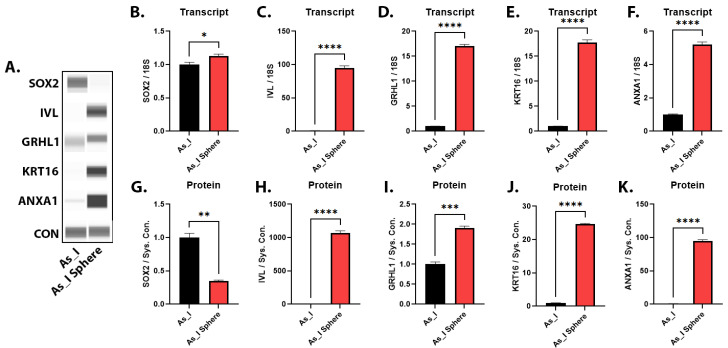
Gene and protein expression in urospheres derived from UROtsa As_I cells. As_I cells were grown as monolayer in regular T-75 flasks (As_I, black bars) or ultra-low attachment T-75 flasks (As_I spheres, red bars) in serum free media. (**A**) Western blot images for As_I cells and As_I spheres. The transcript level for (**B**) SOX2, (**C**) IVL, (**D**) GRHL1, (**E**) KRT16, and (**F**) ANXA1 in As_1 cells and As_I spheres. The protein quantification for (**G**) SOX2, (**H**) IVL, (**I**) GRHL1, (**J**) KRT16, and (**K**) ANXA1 in As_1 cells and As_I spheres. All data is plotted as fold-change compared to the As_I cells. Gene expression was normalized to the 18S housekeeping gene and protein expression was normalized to the Jess system control. The gene measurements were performed in triplicates while protein measurements were performed in duplicates and the values reported are mean ± SEM. A *t*-test was performed to determine significance. Asterisks indicate significant differences between the As_I cells and As_I spheres (* *p* < 0.05, ** *p* ≤ 0.01, *** *p* ≤ 0.001, **** *p* ≤ 0.0001).

**Figure 10 ijms-24-09149-f010:**
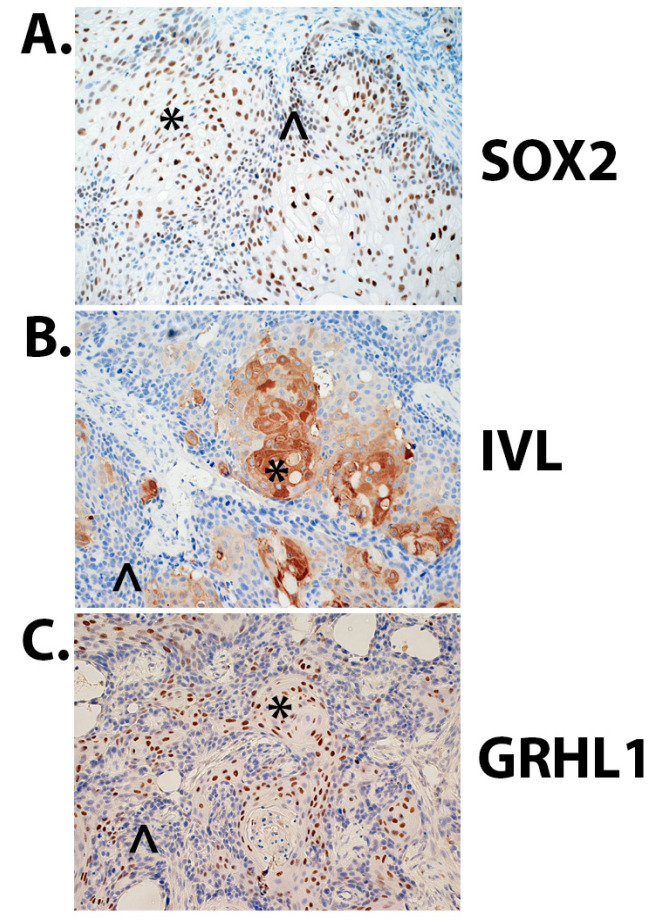
Immunohistochemical analysis of tumor heterotransplants derived from UROtsa As_I cells. (**A**) Nuclear SOX2 expression in the tumors produced by the As_I cells, SOX2 was expressed both in the well-differentiated areas (squamous, *) and the non-differentiated areas (basal, ^). (**B**) Cytoplasmic IVL expression, IVL appears to be mainly expressed in the well-differentiated/squamous areas (*) and mostly absent in basal areas (^). (**C**) Nuclear GRHL1 expression, GRHL1 appears to be mainly expressed in the well-differentiated/squamous areas (*) and mostly absent in basal areas (^). The brown color indicates the presence of the protein whereas the blue/purple color indicates the nuclei that are stained with the counterstain hematoxylin. All images are at a magnification of 200×.

**Table 1 ijms-24-09149-t001:** Immunohistochemical analysis of tumor heterotransplants derived from URO As_I cells.

Protein	INT	%
SOX2	2+	80
IVL	3+	30
GRHL1	3+	25

INT; intensity of staining, %; % of cells staining for a marker, 3+; strong staining, 2+; moderate staining, 1+; weak staining.

## Data Availability

Not applicable.

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
