# Peer review of "Pevonedistat Inhibits SOX2 Expression and Sphere Formation but Also Drives the Induction of Terminal Differentiation Markers and Apoptosis within Arsenite-Transformed Urothelial Cells"

_ijms, 2023, doi:10.3390/ijms24119149_

Round 1

Reviewer 1 Report

Mehus et al. have initially identified the role of SOX2 in promoting proliferation of urothelial cancer (UC) and causing cisplatin (CIS) resistance in arenite-transformed UROtsa cell lines. They further studied the effect of Pevonedistat (PVD), and CIS, alone and in-combination on terminal differentiation markers in UC cells. Combination of PVD and CIS significantly enhanced terminal differentiation markers that may further prevent metastasis.

Here are my recommendations to improve the quality of your manuscript.

1.     The authors are requested to check for grammatical and formatting errors throughout the manuscript.

2.     Page 2 line 48-49: The authors have mentioned that relapse rate for urothelial cancer (UC) is high. How high is it? It would be better if authors could provide the numbers and cite appropriate references.

3.     Page 2: Authors have mentioned about both arsenic and arsenite. For improved understanding of readers, authors are requested to briefly mention about both arsenic and arsenite in terms of their toxicity, solubility, and excretory efficacies.

4.     Page 2 para 3: Apart from Pevonedistat (PVD), are there any other SOX2 inhibitors that have been used in UC studies?

5.     The authors have well written the entire study and discussion. However, they can also seek to provide an overview of the mechanism by which PVD inhibits SOX2.

6.     The authors are requested to mention the future plan of the study, if any.

7.     An abbreviation list covering all the abbreviations and their full forms added in the beginning or at the end of manuscript could simplify the reading process.

8.     In the discussion, authors have discussed about Cancer stem cells (CSCs), some more should be discussed as published in the recent study https://doi.org/10.1016/j.lfs.2021.119465

Some minor changes are requires as suggested in comments.

Reviewer 2 Report

The authors have done a good job of comprehensively characterizing mechanism of action of a small molecular inhibitor of SOX2 and CIS and the effects on bladder cancer cells. The authors have used a range of techniques including both genomic and proteomic metrics. 

The authors have demonstrated the effectivity of their model in showing As3+ mediated SOX2 overexpression as a baseline for comparison. As well, they have evaluated the individual effects of treatments PVD and CIS as well as the combined effects of the treatments on these cell types, mainly in reference to reduced stemness vs semi-squamitization. 

Some general questions:

1) How are the two drugs PVD and CIS typically administered, and is there any dose-dependent mechanism of action? The authors mention that there were some concentrations chosen based on some preliminary studies; can they provide these in supporting information? What are the doses? 

2) Were any in vivo studies conducted that evaluated the effects of PVD and CIS administration in animals that obtained the cancer cells? 

3) Cancer cells tend to be a very heterogenous population. Did the authors look into the single-cell sequencing of genes and if there was any spatial or temporal distribution of any specific genes over the time of culture of through the colonies. 

Reviewer 3 Report

1.       Why use proteomics rather than transcriptomics? Please explain the reason based on experimental design and technical concerns.

2.       As_I, II, III are considered technical or biological repeats?

3.       How do you decide the concentration of PVD used in Fig.3, preliminary data or previous findings, how do you reject the possibilities of cytotoxicity with higher dosage?

4.       Hard to tell the white “cloudy” areas by BF photos with low magnifications, would be helpful to put high magnification would be helpful

5.       The Y axis of Fig. 3N should be 0-100%

6.       To indicate “irregular in shape” or “not tightly packed together”, it would be better to use FIJI measurement to analyze the shape, and use nuclei staining on spheres to show that

7.       Again, you need some higher magnification and resolution in Fig. 6 corresponding to your description in the results (and arrowhead)

8.       Graph 7b does not significantly show what you mentioned in the results

9.       Compare Fig. 8A and 8B, the protein level of SOX2 is higher in parent cells than in treated cells, please explain.

10.   In Fig. 10, would the immunohistochemical analysis result be more convincing if you include tumor hetero-transplants derived from parent cells result and make a comparison?

The conclusive sentences in each paragraph of the results are not based on most cases' findings with over interpretation. Consider rephrasing them and using more accurate words. 
